# Assessment of Psychopathology in Adolescents with Insulin-Dependent Diabetes (IDD) and the Impact on Treatment Management

**DOI:** 10.3390/children8050414

**Published:** 2021-05-19

**Authors:** Maria Melania Lica, Annamaria Papai, Andreea Salcudean, Maria Crainic, Cristina Georgeta Covaciu, Adriana Mihai

**Affiliations:** 1Department of Psychiatry, Faculty of Medicine, George Emil Palade University of Medicine, Pharmacy, Science and Technology of Tirgu Mures, 540139 Targu Mures, Romania; melaniacozma76@gmail.com (M.M.L.); annamariaporkolab@yahoo.com (A.P.); andreea.salcudean@yahoo.com (A.S.); maria.crainic1992@yahoo.com (M.C.); 2Clinical Emergency Hospital for Children‒Child and Adolescent Psychiatry, 400000 Cluj Napoca, Romania; ippdms@yahoo.com; 3IPPD Institute of Psychotherapy and Personal Development, 540044 Tirgu Mures, Romania

**Keywords:** adolescents with insulin-dependent diabetes, treatment adherence, glycosylated hemoglobin, psychopathology

## Abstract

Assessing mental health in children and adolescents with insulin-dependent diabetes (IDD) is an issue that is underperformed in clinical practice and outpatient clinics. The evaluation of their thoughts, emotions and behaviors has an important role in understanding the interaction between the individual and the disease, the factors that can influence this interaction, as well as the effective methods of intervention. The aim of this study is to identify psychopathology in adolescents with diabetes and the impact on treatment management. A total of 54 adolescents with IDD and 52 adolescents without diabetes, aged 12–18 years, completed APS–SF (Adolescent Psychopathology Scale–Short Form) for the evaluation of psychopathology and adjustment problems. There were no significant differences between adolescents with diabetes and control group regarding psychopathology. Between adolescents with good treatment adherence (HbA1c < 7.6) and those with low treatment adherence (HbA1c > 7.6), significant differences were found. In addition, results showed higher scores in girls compared with boys with IDD with regard to anxiety (GAD), Major Depression (DEP), Post-Traumatic Stress Disorder (PTSD), Eating Disturbance (EAT), Suicide (SUI) and Interpersonal Problems (IPP). No significant differences were found regarding the duration of the disease. Strategies such as maladaptive coping, passivity, distorted conception of the self and the surrounding world and using the negative problem-solving strategies of non-involvement and abandonment had positive correlation with poor glycemic control (bad management of the disease). The study highlighted the importance of promoting mental health in insulin-dependent diabetes management.

## 1. Introduction

Insulin-dependent diabetes (IDD) is one of the most common chronic diseases in children and adolescents. The European Association for the Study of Diabetes (EASD) data show that there are over 580,000 children and adolescents diagnosed with IDD in the world [1]. In Romania, in 2017, there were 5516 patients with diabetes aged between 0 and 18 years old; 4989 had IDD and 527 had non-insulin-dependent diabetes [2].

IDD management involves blood sugar measuring, carbohydrate calculation, dosing and administration of insulin, a healthy lifestyle, regular physical activity, rest and avoidance of risky behaviors such as use of alcohol, tobacco and drugs [3]. Compliance with these conditions has a determining role in physical and psychological integrity. Individuals react differently to the disease and treatment requirements [4,5], and psychological processes seem to be involved. The negative influence of psychiatric co-morbidities, such as anxiety, depression and eating disorders, on the management of diabetes is highlighted in several studies [4,6,7], and diabetes, as a chronic illness, is considered one of the main causes of emotional exhaustion and suicide risk factor in children and adolescents with diabetes [8,9]. However, this data does not exclude the existence of mental disorders in children without diabetes. William Dikel says that one in five adolescents suffers from a mental disorder, and the age of adolescence is a vulnerable stage; adults must pay special attention to the manifestations of such disorders at during this time, promoting efficient coping methods [10].

According to Anderson, anxiety and post-traumatic stress disorder are experienced by patients with diabetes [11]. They often can be underdiagnosed due to the similarities between the vegetative symptoms of anxiety and hyperglycemia [12]. Recent research remarked on the existence of eating disorders (binge eating, bulimia nervosa) in adolescents with diabetes, especially in girls [13,14]. There is a relationship between eating disorders and low glycemic control [15]. Diabulimia, an eating disorder specific to patients with IDD, involves individuals deliberately giving themselves less insulin than they need with the goal of losing weight and can result in diabetic ketoacidosis, retinopathy and nephropathy [16].

Chronic disease interferes with normal functionality; patients with severe chronic pathology having an important disability. Continuous therapeutic requirements, the limits imposed by the diet, and the reduction of physical capacities interfere with quality of life and social integration in adolescents with IDD [17]. Many factors influence the image of the disease in terms of a new self-identity and the feeling of efficiency [18,19]. A positive attitude of individuals towards problems increases the possibility of solving them. On the contrary, a negative, catastrophic perception induces a feeling of helplessness, and the chances of solving problems decrease [17,18,19]. Bury believes that chronic diseases are responsible for the disruption of identity, with many psychological repercussions, and depression and anxiety occur when people experience feelings of inefficiency [20].

Studies [21,22] have shown that screening for mental disorders in children and adolescents with diabetes and early interventions can significantly reduce negative effects on disease management and the costs to the healthcare systems.

The aim of this study is to identify and assess psychiatric disorders in adolescents with IDD and the relation to the disease management. The hypotheses of the study are as follows: (1) There are no significant differences in the prevalence of psychiatric pathology among adolescents with IDD and those without IDD (control group). (2) The adolescents with IDD who have poor glycemic control have higher levels of psychopathology compared with adolescents with good glycemic control.

## 2. Material and Methods

This is a prospective pair match control group study on adolescent patients with insulin-dependent diabetes (IDD) between 12–18 years old, compared with control pair match volunteers without diabetes. In this study, 60 adolescent patients with IDD were included; 6 were excluded because of incomplete data. A total of 54 completed the study, and 54 adolescent pair match volunteers without diabetes were recruited, 2 of which were excluded because of incomplete data.

Patients with IDD were recruited consecutively from an outpatient setting for patients with diabetes until the recruitment target was fulfilled. The inclusion criteria were as follows: definitive diagnosis of insulin-dependent diabetes (IDD) after at least 6 months from onset, diagnosis made by a diabetes specialist, patients age between 12 and 18 years old, informed consent of study participation from parents, and adolescents assent to study participation. The exclusion criteria were previous psychiatric diagnosis and/or treatment, incapacity to understand the test demands, incomplete data on tests, and lack of laboratory results in last month (glycosylated hemoglobin-HbA1c). The control group included volunteer adolescents without diabetes and/or psychiatric diagnosis or treatment, recruited in the same period of time as pair matches (for gender, age, level of education and rural/urban setting). The exclusion for psychiatric history (previous psychiatric diagnostic and/or treatment) for both groups was done based on self-reporting from adolescents and their parents. Informed consent and assent was obtained from all participants and from their parents. All data were coded because of confidentiality issues. The study was conducted in accordance with the Declaration of Helsinki, and the protocol was approved by the Ethics Committee of George Emil Palade University of Medicine Pharmacy Science and Technology (UMFST) Targu Mures, Romania(nr. 941/26.05.2020). All adolescents completed APS–SF (Adolescent Psychopathology Scale–Short Form) for the evaluation of common psychopathology and adjustment problems in ages 12–18 years (pen-and-paper tests). The total duration of the psychological evaluation was 20 min. The scoring was done by computer (PC Based Software; PARiConnect).

The APS–SF (Reynold, 2000) [23] is a multi-dimensional instrument that measures psychopathology and personality characteristics and their severity. Derived from the Adolescent Psychopathology Scale (APS), it was validated on the Romanian population [24]. The validity of the scale results from high correlation with other disorder assessment tools [24]. It consists of 115 items in 12 clinical scales and 2 validity scales. The items were designed in correspondence with the clinical symptoms of DSM-IV, axis 1, and with characteristics of the five personality disorders from axis 2 of DSM-IV, also evaluating other emotional issues and problematic behavioral manifestations in adolescence. The APS–SF clinical scales include Conduct Disorder (CND), Major Depression (DEP), Post-Traumatic Stress Disorder (PTSD), Eating Disturbance (EAT), Academic Problems (ADP), Self-Concept (SCP), Oppositional Defiant Disorder (ODD), Generalized Anxiety Disorder (GAD), Substance Abuse Disorder (SUB), Suicide (SUI), Anger/Violence Proneness (AVP), and Interpersonal Problems (IPP). The APS–SF validity scales include Defensiveness (DEF) and Consistency Response (CNR).

For the statistical analysis, the data from the 14 scales were processed quantitatively with IBM SPSS Statistical 20; the descriptive analyses included means, correlations, independent-Sample T Test, ANOVA simple, and d-Cohen.

### 3. Results

The present research attempted an analysis of psychological functioning through the self-assessment of behavior, observation of rules, predisposition to react through anger/violence, perception of school problems, eating disorders, interpersonal problems, anxiety, depression, post-traumatic stress and ideation, as well as suicide attempts in adolescents with insulin-dependent diabetes (IDD) (*n* = 54) and those without this chronic disease (*n* = 52). Demographical characteristics are shown in Table 1.

There were no significant differences in the median distribution of psychiatric psychopathology identified by the ASP–SF between two groups—adolescents with and without diabetes (Figure 1). The first hypothesis, according to which there are no significant differences in the prevalence of psychiatric pathology among adolescents with IDD and those without IDD (group control), was thus accepted.

For checking the second hypothesis, according to which the adolescents with IDD with poor glycemic control have higher levels of psychopathology compared with adolescents with good glycemic control, we divided the group of patients with IDD into two groups: one for good glycemic control and the second for poor glycemic control. The criterion for grouping adolescents with IDD was glycosylated hemoglobin HbA1c value, which is considered to be the most important indicator of glycemic control. HbA1c < 7.6 was considered good glycemic control and HbA1c > 7.6 was considered poor glycemic control, as recommended by the American Diabetes Association (ADA) and ISPAD (International Society for Pediatric and Adolescent Diabetes) [25,26].There were significant differences reflected in the scores of all subscales between adolescents with diabetes with poor glycemic control and those with good glycemic control. For Conduct Disorder (CND), Oppositional Defiant Disorder (ODD), Substance Abuse Disorder (SUB), Anger/Violence Proneness (AVP), Academic Problems (ADP), Generalized Anxiety Disorder (GAD), Post-Traumatic Stress Disorder (PTSD), Major Depression (DEP), Eating Disturbance (EAT), Suicide (SUI), Self-Concept (SCP), and Interpersonal Problems (IPP), the levels were higher in adolescents with high levels of HbA1C (poor glycemic control). They obtained, on average, significantly higher scores on the APS–SF compared to adolescents with a good glycemic control. The differences are statistically significant for all psychological problems. The research data sustain the second hypothesis, according to which there is a significant association between poor glycemic control and psychiatric psychopathology (Figure 2 and Table 2).

The correlation between HbA1c value and the variables constituted by APS–SF subscales was determined using the Spearman’s test, which demonstrated the existence of a strong positive correlation (*p* = 0.001) between almost all the subscales and HbA1c (coefficient value > 0.50) and a medium correlation between EAT subscale and HbA1c (rs > 0.30). (Table 3)

Another variable taken in consideration from existing data was the duration of the disease (calculated in months), dividing IDD patients group into three subgroups: A.from 10 to 30 months (*n* = 13)B.from 31 to 60 months (*n* = 17)C.>60 months (*n* = 24)

For multiple comparison and analysis of variance, we used ANOVA Simple. There were no significant differences between subgroups regarding APS–SF subscale scores. No differences were found when we used multiple variances Bonferroni and Tamhane. The higher score was indicated for group B (31–60 months) at the Generalized Anxiety Disorder subscale (m = 7.47) (Figure 3).

The last independent variable observed in this study was gender: Female (F) *n*= 21 and Male (M) *n*= 33 (Table 4). 

In order to verify whether the factors of the APS–SF scale for adolescents with IDD differed according to gender, we applied the t test for independent samples. The results were statistically significant; there were significant differences between female and male participants at the following subscales: GAD, PTSD, DEP, EAT, SUI, IPP.

### 4. Discussion

Previous studies in the field have shown that patients with diabetes are much more prone to various psychiatric disorders than healthy people, with the emphasis being on the presence of the disease and less on its management status.

There are no significant differences in the media distribution of mental disorders identified by ASP–SF subscales between adolescents with IDD and those without diabetes, but high levels of psychopathology in both groups; this is in contrast with other studies, which found higher risk of mental disorders in adolescents with IDD [11].

There are not many studies in the literature regarding the mental health of adolescents with IDD; however, it was proven that anxiety, depression, oppositional disorder, self-concept problems, interpersonal problems, and anger and violence proneness, in a certain context such as a disease or other stressors, could influence adaptation and reduce the ability to solve problems or to search for solutions. However, high levels of psychopathology identified in both groups indicate communication problems, conflicts, sadness, maladaptive coping strategies, inefficient educational principles, lack of learning strategies, distorted or irrational thinking, and lack of school or extracurricular orientation in the sense of facilitating success and increasing self-confidence. The presence and severity of psychopathology in adolescents may indicate a risk of worsening if not treated in a timely and appropriate manner. Psychiatric psychopathology in adolescents with diabetes prevents or hinders the achievement of an optimal glycemic balance, with short-, medium-, and long-term complications, leading in many cases to vision loss, kidney failure, strokes, amputations, severe cognitive impairment, or even death. In addition, the management of the disease and the difficulties in achieving glycemic control could influence the psychological well-being of young people and facilitate the onset of depression, anxiety, communication and relationship difficulties as well as impaired motivation, attention, and memory, which are important in achieving academic success. At this age, eating restrictions, meal and treatment schedules, and parental control can be perceived as obstacles to personal autonomy and development. Young people with IDD who have eating disorders have significant difficulties in managing blood glucose and insulin doses; at the same time, difficulties in glycemic control or weight control could lead to maladaptive nutritional measures, such as restrictive diets, vomiting, or insulin dose reduction [27]. Interpersonal problems are generally reflected in the avoidance of or faulty relationships with medical staff, family members, colleagues, and teachers and are generally associated with low adherence to treatment, isolation, academic problems, eating disorders, and depression. In addition, negative social perceptions related to the disease can have emotional effects and affect interpersonal relationships as well as social and school adaptation. Even a relation of association could be supported for a causal relationship; other additional variables should be evaluated (environmental, familial, and genetic factors). Several mental disorders in adolescents have hereditary transmissions of vulnerability, but the onset of symptoms depends on stressful environmental conditions, social situations, parenting style, and disturbed family relationships [28,29,30,31].

Due to the need for continuous self-monitoring, repeated injections, restrictive diets, and mandatory medical check-ups, adolescents with diabetes may experience specific distress [32]; failures to achieve a glycemic balance can be a source of frustration and guilt. Some of the children and adolescents with diabetes resort to maladaptive coping strategies, anxiety about the disease, anger and aggression, and alcohol consumption [33,34], which were highlighted in adolescents with high levels of HbA1c from our study.

Although, major depressive disorder is considered to be closely related to diabetes, Gonzales emphasizes the importance of distinguishing between this and the specific distress caused by the disease [32]. However, in a study conducted in 2011, he showed that, although the incidence of depression is higher in adolescents with diabetes than those without diabetes, the differences are not significant, as previous studies have shown. In addition, our study argues that psychopathology is not caused by the disease itself, as some previous studies concluded, resulting in a positive correlation of psychiatric disorders and poor glycemic control. This fact indicates the importance of early identification of those factors (biological, maternal profile, psychological, and social) that produce and maintain psychiatric disorders in adolescents with diabetes as well as the relationship with the management of the disease [35]. In addition, improving supportive relationships and continuously adjusting therapeutic dietetic plans, including tracking food composition, in order to increase glycemic balance, could reduce the risk of developing psychopathology and increase quality of life [36,37]. Regarding adolescents with poor glycemic control and those with good glycemic control, important differences in Anger/Violence Proneness (AVP), Interpersonal Problems (IPP), Major Depression (DEP), Generalized Anxiety Disorder (GAD), Self-Concept (SCP), Post-Traumatic Stress Disorder (PTSD), Oppositional Defiant Disorder (ODD), and Academic Problems (ADP) scale scores were seen in our research. The higher scores can be translated into strong emotional and behavioral imbalance, difficulties in understanding and accepting the disease and the treatment, and difficulties in coping with the distress, conflicts, social isolation, feelings of inefficiency, learning difficulties, and/or low interest in school performance.

Sadness, negative thinking, maladaptive coping, passivity, distorted conception of the self and the surrounding world, using negative problem-solving strategies, and abandonment had positive correlation with poor glycemic control (bad management of the disease) in our study. In addition, as the manual APS–SF [28] indicates, we considered the presence of association between critical items that express both internalization and outsourcing disorders (IPP-AVP; DEP-PTS; DEP-IPP; DEP-ADP; SCP-ADP) [24]. The absence of conduct disorders, suicidal ideations or behavior, substance abuse disorder, and anger violence disorder in adolescents with good glycemic control strengthens the connection to disease management.

In the literature [38], the association between psychological problems and poor glycemic control can influence the prognosis of the chronic disease of insulin-dependent diabetes (IDD). The personal significance of the disease, the expectations, the beliefs about how the adolescent can control the symptoms of the disease and its evolution, and the degree of treatment efficiency are cognitive schemes in a circular relationship with the disease management, both influencing and being determined by it. Poor glycemic control can also be the cause of increasing psychopathology in adolescents.

Treatment compliance alone cannot influence glycemic control, as clinical practice and literature have often highlighted. To explain the bad management of the disease and find functional measures, further investigations are needed. When the care of adolescents with IDD is limited to increase adherence to the therapy, the biological, psychological, and socio-economic conditions and medical system settings are neglected. The results are incomplete, resulting in unrealistic images and an inefficient approach to each case. This approach also increases feelings of self-guilt and may decrease perceived self-efficacy, with the risk of subsequent non-involvement. This must be taken into account both in medical practice and in scientific research.

Another important parameter is C-peptide level, used to measure pancreatic beta cell function (endogenous insulin production). It is also important in personalized management of type 1 diabetes, determining future glycemic control and the risk of diabetes complications (nephropathy, neuropathy, retinopathy, ulcers) [39]. Lower C-peptide values can explain poor glycemic control, glucose variability, and increased HbA1c values.

The available data regarding the elevated PTS scores in adolescents in the Romanian population are limited. The life-threatening events experienced by healthy adolescents in our sample were represented by domestic violence or abandonment; in the IDD adolescent population, they could be represented by severe medical complications such as coma, which could be a life-threatening event. The exposure to conflicts, violence, separation, lack of social support, and insecure environments make this group more vulnerable to negative events because of emotional instability and reactivity. The scale does not evaluate the traumatic event, just whether it has been present in one’s lifetime.

Duration of the IDD (months from the time of diagnosis) does not appear to be related with psychopathology in our study. The diagnosis of chronic disease goes through several phases, until acceptance and adaptation, and during this process, a specific distress may arise. Periodical psychological evaluations, including family interviews, along with medical ones could better indicate the degree of disease management, mental health status, and the quality of life.

From the analysis of gender influence, there are significant scores that indicate a higher prevalence of psychopathology in female adolescents with IDD compared with males. Significant differences were seen in post-traumatic stress, depression, eating disorders, anxiety, interpersonal problems, and suicide subscales. The scores highlight girls’ difficulties in coping with stress, the associated emotional experience, and communication problems.

The strengths of the present study consist of underlining the relationship between psychopathology and diabetes management in adolescents with insulin-dependent diabetes (IDD). Regarding this, in clinical practice, the attention is predominantly focused on laboratory tests and physical symptoms and less on psychological and social aspects. The aim of this study was to transcend these boundaries, including the use of mental health assessment in identifying barriers and resources in IDD management. Psychological evaluation can provide useful indicators in designing both intervention and prevention programs. The APS–SF scale evaluates the psychiatric psychopathology but does not offer a psychiatric diagnostic. The sample size of the study group and the control group was not very large, such that even though an association between psychopathology and glycemic control was proven, a relation of causality cannot be supported in this study. Family characteristics (beliefs, habits, parenting style, socio-economic conditions, and family relationships) and additional clinical and laboratory analyses (e.g., Peptide-C, body mass index) as well as data collected from parents and teachers on adolescent behavior could be included in a future study.

### 5. Conclusions

Adolescents with IDD and those without IDD face psychological problems, almost to the same extent. A significant association was found between high levels of psychopathology and poor glycemic control. The female adolescent IDD patients were more vulnerable to psychopathology than males.

The results can contribute to an overview of the importance of psychological evaluation in diabetes and its role in establishing efficient strategies in the management of the disease.

## Figures and Tables

**Figure 1 children-08-00414-f001:**
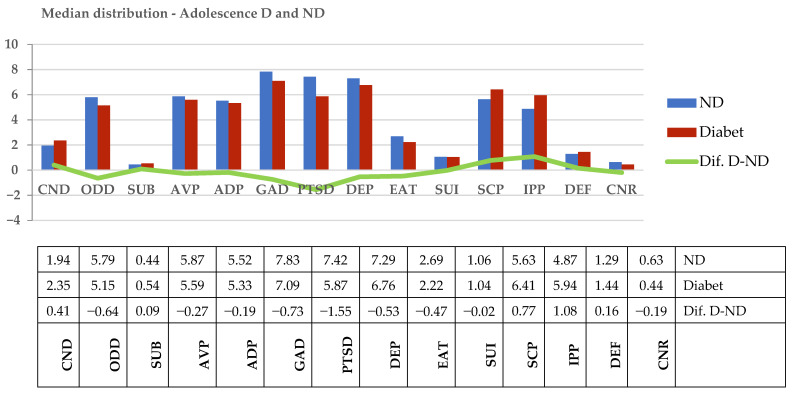
Median distribution of ASP–SF subscales in adolescents with IDD (D) and control group (ND). Legend: Conduct Disorder (CND), Oppositional Defiant Disorder (ODD), Substance Abuse Disorder (SUB), Anger/Violence Proneness (AVP), Academic Problems (ADP), Generalized Anxiety Disorder (GAD), Post-Traumatic Stress Disorder (PTSD), Major Depression (DEP), Eating Disturbance (EAT), Suicide (SUI), Self-Concept (SCP), Interpersonal Problems (IPP), Defensiveness (DEF), and Consistency Response (CNR).

**Figure 2 children-08-00414-f002:**
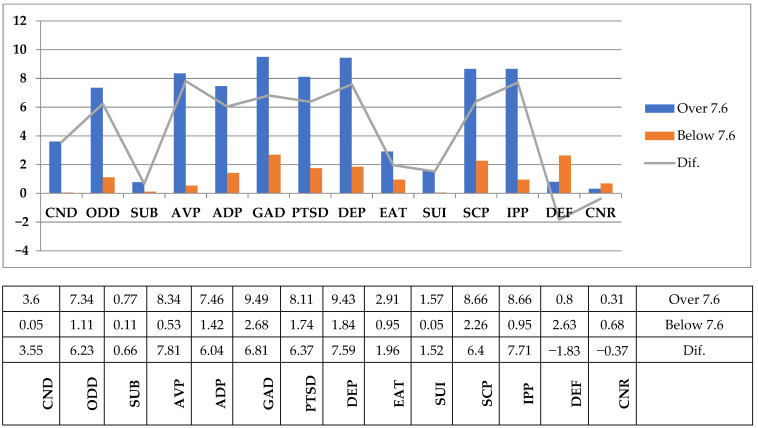
Media distribution of ASP–SF subscales in adolescents with IDD and high and low glycemic control. Legend: Conduct Disorder (CND), Oppositional Defiant Disorder (ODD), Substance Abuse Disorder (SUB), Anger/Violence Proneness (AVP), Academic Problems (ADP), Generalized Anxiety Disorder (GAD), Post-Traumatic Stress Disorder (PTSD), Major Depression (DEP), Eating Disturbance (EAT), Suicide (SUI), Self-Concept (SCP), Interpersonal Problems (IPP), Defensiveness (DEF), and Consistency Response (CNR).

**Figure 3 children-08-00414-f003:**
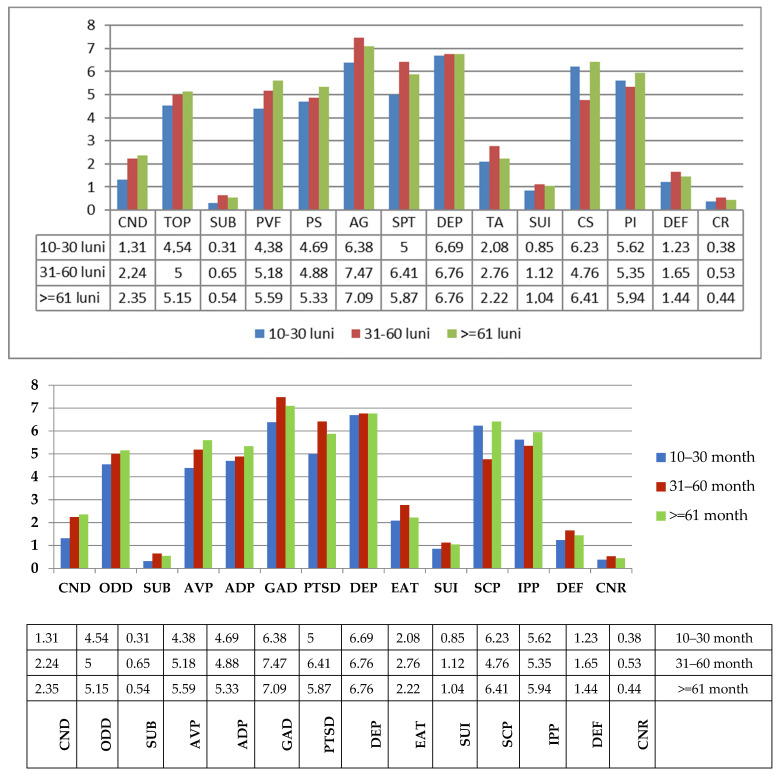
Comparison between A, B, C subgroups concerning the duration of disease. Legend: Conduct Disorder (CND), Oppositional Defiant Disorder (ODD), Substance Abuse Disorder (SUB), Anger/Violence Proneness (AVP), Academic Problems (ADP), Generalized Anxiety Disorder (GAD), Post-Traumatic Stress Disorder (PTSD), Major Depression (DEP), Eating Disturbance (EAT), Suicide (SUI), Self-Concept (SCP), and Interpersonal Problems (IPP), Defensiveness (DEF) and Consistency Response (CNR).

**Table 1 children-08-00414-t001:** Demographic characteristics of sample.

Demographic Characteristics	IDD Patients (*n* = 54)	Control Group (*n* = 52)	*p*
Mean/SD (Age)	15.28/1.774	14.38/1.619	
Early stage (10–14 years old)	37% (*n* = 20)	53.8% (*n* = 28)	*p* =0.08
Middle stage (15–17 years old)	51.9% (*n* = 28)	40.4% (*n* = 21)	*p* = 0.23
Late adolescence (18–24 years old)	11.1% (*n* = 6)	5.8% (*n* = 3)	*p* = 0.32
Gender			
-Female	38.9% (*n* = 21)	50% (*n* = 26)	*p* = 0.24
-Male	61.1% (*n* = 33)	50% (*n* = 26)	*p* = 0.24
Urban/rural			
-Urban	98.1% (*n* = 53)	100% (*n* = 52)	*p* = 0.32
-Rural	1.9% (*n* = 1)	0% (*n*= 0)	*p* = 0.32
Level of education			
-Primary school	51.9% (*n* = 28)	61.5% (*n* = 32)	*p* = 0.31
-High school	48.1% (*n* = 26)	38.5% (*n* = 20)	*p* = 0.31

**Table 2 children-08-00414-t002:** Significant association between adolescents with psychiatric psychopathology and poor glycemic control.

Variable	HbA1c > 7.6 m 1	HbA1c < 7.6 m 2	t	df	Sig 2 Tailed	d-Cohen
CND	3.60	0.05	7.47	34.84	0.001	2.65
ODD	7.34	1.11	8.42	46.99	0.001	2.57
SUB	0.77	0.11	3.61	44.71	0.001	1.13
AVP	8.34	0.53	7.48	37.87	0.001	2.54
ADP	7.46	1.42	8.60	49.06	0.001	2.57
GAD	9.49	2.68	6.57	51.95	0.001	1.90
PTSD	8.11	1.74	6.05	51.70	0.001	1.76
DEP	9.43	1.84	6.44	45.72	0.001	1.99
EAT	2.91	0.95	2.61	51.91	0.012	0.75
SUI	1.57	0.05	4.67	35.80	0.001	1.63
SCP	8.66	2.26	6.91	51.75	0.001	2.01
IPP	8.66	0.95	7.08	48.31	0.001	2.13
DEF	0.80	2.63	−5.17	27.12	0.001	2.07
CNR	0.31	0.68	−1.60	22.95	0.123	0.69

Legend: Conduct Disorder (CND), Oppositional Defiant Disorder (ODD), Substance Abuse Disorder (SUB), Anger/Violence Proneness (AVP), Academic Problems (ADP), Generalized Anxiety Disorder (GAD), Post-Traumatic Stress Disorder (PTSD), Major Depression (DEP), Eating Disturbance (EAT), Suicide (SUI), Self-Concept (SCP), Interpersonal Problems (IPP), Defensiveness (DEF), and Consistency Response (CNR).

**Table 3 children-08-00414-t003:** Correlation between HbA1c value and the APS–SF subscales (Spearman’s rho).

	Variables	M	SD	Correlation
1	HbA1C	9.03	2.401	-
2	CND	2.35	2.816	813 **
3	ODD	5.15	4.436	860 **
4	SUB	0.54	0.84	590 **
5	AVP	5.59	6.141	851 **
6	ADP	5.33	4.207	834 **
7	GAD	7.09	5.349	755 **
8	PTSD	5.87	5.370	739 **
9	DEP	6.76	6.381	766 **
10	EAT	2.22	3.214	364 **
11	SUI	1.04	1.693	712 **
12	SCP	6.41	4.939	792 **
13	IPP	5.94	6.026	830 **

** Correlation is significant at the 0.01 level (2-tailed).

**Table 4 children-08-00414-t004:** Significant results for female vs. male (adolescents with diabetes).

Dependent Variable	Mean Female	Mean Male	t	df.	Sig.	D
GAD	9.48	5.58	2.512	30.140	0.018	0.938
PTSD	8.71	4.06	3.092	30.522	0.004	1.148
DEP	9.43	5.06	2.335	30.022	0.026	0.874
EAT	4.86	0.55	5.283	22.972	0.000	2.261
SUI	1.71	0.61	2.100	24.929	0.046	0.862
IPP	8.24	4.48	2.098	29.866	0.044	0.787

Legend: Generalized Anxiety Disorder (GAD), Post-Traumatic Stress Disorder (PTSD), Major Depression (DEP), Eating Disturbance (EAT), Suicide (SUI), and Interpersonal Problems (IPP).

## Data Availability

Data are available on request (due to privacy and ethical restrictions) from the corresponding autor (A.M.).

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
