# Peer review of "Assessment of Psychopathology in Adolescents with Insulin-Dependent Diabetes (IDD) and the Impact on Treatment Management"

_children, 2021, doi:10.3390/children8050414_

Round 1
Reviewer 1 Report
The Authors try to find association about psychopathology and emotional stress between IDD and control group.
The significant finding is psychopathology and treatment adherence. However, the case number is limited and the confounding factors were too many.
The introduction and discussion sections including many references to support the theory of the psychopathology. However, these references were too many and redundant. The discussion section should be concise and focus on the findings of this manuscript.
However, this manuscript contributed to understand the relationship of psychopathology and diabetes management in adolescents, and may provide further therapeutic success and improve quality of life of young patients with IDD.
Here are some concerns:
The introduction and discussion sections should be re-write to improve reading and emphasize the findings.
The HbA1c was cut by 7.6%, could the lower part of patients experienced major hypoglycemic event? Could you trisection the HbA1c?
What’s the IRB number?
The table 1, age column, should be re-arranged. The alignment is not clear.
Discussion section:
Paragraph 1 and 2: not related to the results.
This section is too long and not very relevant to the results.
Reviewer 2 Report
I totally disagree that the differences between adequately controlled and poorly controlled type 1 diabetes is due to noncompliance.The more likely reason is that the higher glucose levels are the cause of increased psychopathology and that factors such as social situations most likely account for differences.In addition due to lower preserved beta cell function many type 1 patients are more brittle.This could be easily documented with C-peptide levels.Previously school preformance has been related to HbA1c and this should be included in the text.A conclusion needs to be added to the Abstract.
Round 2
Reviewer 2 Report
Appreciate the changes you have made. The changes were not enough to warrant publication.The concept that poor glycemic control was due to non-compliance was NOT corrected.
Round 3
Reviewer 2 Report
After reviewing the current manuscript which has addressed my previous comments,I change my recomendation to ACCEPT